# The Impact of High-Pressure Processing on Physicochemical Properties and Sensory Characteristics of Three Different Lamb Meat Cuts

**DOI:** 10.3390/molecules25112665

**Published:** 2020-06-08

**Authors:** Qianli Ma, Nazimah Hamid, Indrawati Oey, Kevin Kantono, Mustafa Farouk

**Affiliations:** 1Department of Food Science, Faculty of Health and Environmental Sciences, Auckland University of Technology, Auckland 1010, New Zealand; mql081228@hotmail.com (Q.M.); kkantono@aut.ac.nz (K.K.); 2Department of Food Science, University of Otago, Dunedin 9054, New Zealand; indrawati.oey@otago.ac.nz; 3Riddet Institute, Massey University, Palmerston North 4474, New Zealand; 4AgResearch MIRINZ, Ruakura Research Centre, Hamilton 3240, New Zealand; mustafa.farouk@agresearch.co.nz

**Keywords:** high pressure processing, fatty acids, free amino acids, oxidative stability, lamb, non-thermal processing

## Abstract

This study investigated the effects of high pressure processing (HPP) on the physicochemical properties and sensory characteristics of different lamb meat cuts. Lamb meat discolouration occurred when HPP was applied at 400 and 600 MPa. Thiobarbituric acid reactive substances (TBARS) values significantly increased with pressure increase from 200 to 600 MPa for loin cut, and 300 to 600 MPa for shoulder and shank cuts. Saturated fatty acid and polyunsaturated fatty acid content significantly decreased with pressure increase from 200 to 600 MPa for shank and shoulder cuts, and 300 to 600 MPa for loin cut. Free amino acids content significantly increased in shank and loin cuts with pressure increase after 200 MPa, and in shoulder cuts after 400 MPa. In addition, samples treated with HPP at high pressure levels of 400 and 600 MPa were associated with browned, livery and oxidized flavours. The pressure levels applied and type of cuts used are important considerations during HPP processing as they influenced physicochemical and sensory properties of lamb samples.

## 1. Introduction

Consumer demand for products that are fresh tasting, additive-free, microbiologically safe, convenient to use and shelf stable is increasing. Raso and Barbosa-Cánovas [1] identified that an ideal processing method should be able to inactivate spoilage and pathogenic microorganisms, reduce degradation of organoleptic and nutritional properties, and produce an acceptable product for consumers that meets the standards of regulatory agencies. Although chilling and freezing can maintain a certain degree of food freshness, growth of microorganisms is commonly delayed or inhibited. Thermal processing, on the other hand, may inactivate microorganisms and enzymes but in turn can adversely affect the sensory qualities of the final product such as appearance, taste and flavour as well as its nutritional value [2].

High pressure processing (HPP) is an industrially tested technology that offers a more natural and environmentally friendly alternative for shelf life extension of food products [3]. Although HPP is considered a “non-thermal process”, its use in high-fat foods causes a significant increase in oxidative processes such as lipid oxidation in minced beef [4], ham [5], turkey thigh muscles [6], chicken breast muscles [7], and abalone [8]. The use of HPP with meat has been somewhat limited as various pressure and temperature levels required to tenderise post-rigor meat may denature myoglobin that subsequently leads to an unacceptable meat colour [9]. Studies have reported that HPP pressurization of meat can result in significant colour changes in fresh meat colour of chicken breast fillet [10], beef [11], pork [12], and lamb [13]. With lipid oxidation induced by HPP, some changes in individual fatty acids may occur. McArdle et al. [13] reported that high pressure had no effect on polyunsaturated/saturated fatty acid (PUFA/SFA) or omega 6/omega 3 (n6/n3) ratio of beef (*M. pectoralis profundus)* muscle, but had a significant effect on the sum of saturated (SFA), monounsaturated (MUFA) and polyunsaturated (PUFA) fatty acids. In contrast, the PUFA/SFA ratios of pressurized lamb (*M. pectoralis profundus*) muscles were significantly higher compared to control samples, but without significant effects on omega 6/omega 3 (*n*6/*n*3) ratios.

HPP can also modulate the proteolytic activities of meat and influence quality. Ohmori et al. [14] reported that the free amino acids content of beef rounds significantly increased when meat was treated at 100–400 MPa compared to control samples. However, another research established no significant differences in the amino acids and peptide content of beef [15] with HPP treatment. HPP was however found to influence the flavour, juiciness and aroma of chicken breast fillet.

HPP can influence sensory properties and consumer acceptance of meat. Kruk et al. [16] reported that treatment at 300 MPa significantly reduced flavour, aroma and juiciness, and treatment at 450 MPa resulted in the weakest aroma. Rodríguez-Calleja et al. [10] further found that pressurized chicken breast fillet at 300 MPa for 5 min was more acceptable, and had more chicken aroma attribute compared to control. Morton et al. [16] further demonstrated that beef (*M. longissimus thoracis*) treated at 175 MPa did not influence the juiciness and flavour of meat but significantly improved the overall acceptability and eating scores. Temporal Dominance of Sensations (TDS) has been recently used to understand the changes in the sensory perception of meat during consumption. Ma et al. [17] evaluated the effects of chilling and freezing prior to pulsed electric field processing (PEF) on volatile profile and sensory attributes of different cooked lamb muscles (i.e., shoulder, rib and loin). The TDS results showed that both storage and PEF treatment affected the temporal flavour of meaty and oxidized flavour attributes. Particularly, longer storage period was associated with oxidized flavour, while PEF treated samples were associated with browned, juicy, livery, and meaty flavour attributes. In another study by Kantono et al. [18], TDS results showed that although oxidation was the most dominant temporal perception, the samples were only found to be detrimental to the sensory quality of PEF processed beef muscles stored for 7 days.

It is evident from previous studies that non-thermal processing can influence the physicochemical and sensory properties of meat. Hence, the objective of this study was to evaluate the effect of HPP treatment on three different lamb meat cuts (shank, loin and shoulder) in terms of colour, lipid oxidation, fatty acids, free amino acids, and temporal sensory changes using the TDS methodology to build upon the growing body of knowledge within the area of non-thermal technology applications on meat.

## 2. Materials and Methods

### 2.1. Preparation of Meat Samples and HPP Processing

The skeletal muscles (i.e., shoulder (*Musculus biceps brachii*), loin (*Musculus longissmus*), and shank (*Musculus infraspinatus*)) used in this study were obtained from six different lambs (cold carcasses weight of 140.5–150.5 kg) at 48 h post-mortem. The research material, including the animals, carcass cutting, packaging and freezing, were conducted at the AgResearch research facility (Hamilton, New Zealand). Each muscle was divided into five separate blocks, vacuum packed in polyethylene plastic bags with following labels: 1) Control, 2) HPP-200, 3) HPP-300, 4) HPP-400, and 5) HPP-600, and immediately frozen at −18 °C in a digitally temperature-controlled thermostat freezer with static flow cold air temperature of −20 °C prior to HPP processing. Samples were then thawed overnight at 4 °C prior to HPP treatment. In summary, three different muscles (*n*_muscle_ = 3) were divided into five separate parts (*n*_parts_ = 5) from a sample population of six lambs (*n*_lambs_ = 6). Therefore, a total of 90 samples were retrieved and their physicochemical properties were analysed in triplicates. Samples were placed in a commercial −20 °C freezer for one week until further analysis.

Pressurization of lamb meat was conducted using an industrial scale HPP equipment (HPP 055, Multivac, Multivac Sepp Haggenmüller GmbH & Co., Wolferschwenden, Germany). Water was used as the pressure-transmitting medium, with the initial temperature around 7–8 °C. The temperature reached after pressure build up was less than 25 °C. The rate of pressure build-up was conducted at 125 MPa/min. Packaged lamb samples were loaded in a cylindrical loading container and HPP-treated at 200, 300, 400, and 500 MPa. Pressure was held for one minute once the targeted pressure was achieved. After depressurisation, all samples were transported and stored at −20 °C for further analysis.

### 2.2. Colour Stability

A Hunter LAB (45/0, Colour flex, Reston, VA, USA) colorimeter was used to measure the meat colour variables in terms of lightness (L*), redness (a*) and yellowness (b*). Illuminant D65 (Daylight at noon) was used with standard 8 mm diameter aperture at 10 ° standard observer similarly described by Kantono et al. [18]. Calibration and standardisation were performed using a ceramic white tile following the instruction manual. The HPP processed meat samples were cut into cubes (2 cm × 2 cm × 2 cm) and placed in a covered petri dish that was placed at the top of the colorimeter lens and L*, a* and b* values recorded. Measurement was done in triplicates for each sample. L* value is measured from black to white, redness is measured from green to red, and b* measured blue to yellowness. In addition, chroma (C*) and hue angle (h*) were then calculated. C* indicates red colour intensity, while h* indicates less red and discolouration [18].

### 2.3. Determination of Lipid Oxidation Using TBARS

The extent of lipid oxidation in meat samples was measured using the thiobarbituric acid reactive substances (TBARS) method as described by Nam and Ahn [19]. Minced meat (5 g) was homogenized with 15 mL of deionized distilled water using a homogenizer (L5M-A Laboratory Mixer, Silverson^®^, East Longmeadow, MA, USA) at 14,000 rpm for 30 s. One millilitre of the meat homogenate was transferred to a test tube and 50 μL of butylated hydroxytoluene (7.2% *w*/*v* in ethanol) and 2 mL of a thiobarbituric acid (TBA)–trichloroacetic acid (TCA) solution (20 mM TBA in 15% (*w*/*v*) TCA) were added. The mixture was vortexed and then incubated in a boiling water bath for 15 min to develop colour. Then samples were subjected to cooling for 10 min, and vortexed before being centrifuged for 15 min at 2500× *g*. The absorbance of the resulting upper layer was measured at 531 nm against a blank prepared with 1 mL deionised water and 2 mL TBA/TCA solution. The amount of TBARS was expressed as milligrams of malondialdehyde per kg of meat. A standard curve was constructed using tetraethoxypropane (ranging from 41.76–62.64 mmol/L).

### 2.4. Determination of Fatty Acids

Fatty acid methyl esters (FAME) were prepared according to Garcia, Hernandez, and Lozano [20]. Fifty milligrams of ground, freeze-dried (48 h) sample was measured into a 4 mL amber vial. A 10 µL aliquot of tridecanoic acid (2 g/L) was added as an internal standard. Toluene (490 µL) and freshly prepared 5% methanolic HCl (750 µL) were then added before filling the vial with nitrogen. After incubation in the water bath at 70 °C (2 h), vials were cooled to room temperature before 6% aqueous K_2_CO_3_ (1 mL) and toluene (500 µL) were added. After centrifugation at 1100× *g* for 5 min, the top layer was removed using a glass Pasteur pipette for FAME analysis.

Derivatized methyl esters of fatty acids were separated and quantified using a Shimadzu GC 2010 with a Flame Ionisation Detector (FID) (Kyoto, Japan). The column used was a 0.25 mm × 30 m × 0.25 µm film thickness, fused-silica column (ZB-WAX, Phenomenex, CA, USA). Nitrogen was used as the carrier gas, with a split ratio of 50, a head pressure of 8.7 PSI, and 1 mL/min column flow. The injector temperature was set at 250 °C. The initial oven temperature was programmed at 140 °C, then increased to 245 °C at 5 °C/min, and held at 245 °C for 15 min. Thirty-seven FAME standards (Supelco product 47885-U) obtained from Sigma-Aldrich (Sydney, Australia) were serially diluted to six concentrations (from 10 to 0.3125 g.L^−1^), and the standard calibration curve constructed was used for quantitative analysis.

### 2.5. Determination of Free Amino Acids

The EZFaast^TM^ amino acid analysis kit (Phenomenex, Macclesfield, UK) was used for the analysis of amino acids using a gas chromatograph fitted with a flame ionization detector (GC–FID) according to manufacturer’s instructions similar to Liu et al. [21]. Meat sample (0.1 g) was mixed with 1 mL of methanol in an Eppendorf tube. The mixture was vortexed and then centrifuged at 2000× *g* for 2 min. The aqueous portion was retained for further analysis. The EZ: faast™ amino acid analysis kit (Phenomenex, Torrance, CA) was used to extract and derivatize the free amino acids in the lamb samples. Samples were analysed using a Shimadzu GC-2010 fitted with a flame ionization detector, and Zebron ZB-AAA GC column (5%-phenyl-95%-dimethylpolysiloxane phase, 30 m × 0.53 mm × 1.50 µm) (Phenomenex). Nitrogen was used as the carrier gas and pressure was set to 60 Pa with a 2.3 mL/min column flow rate. The temperature program started at 40 °C, increased to 110 °C at 50 °C/min, then to 320 °C at 20 °C/min, and held at 320 °C for 2 min. Norvaline (0.2mmol/l) was used as an internal standard, and calibration curves of 17 amino acids (alanine, glycine, threonine, serine, proline, glutamic acid, aspartic acid, valine, leucine, isoleucine, methionine, phenylalanine, lysine, histidine, tyrosine, and tryptophan) were used for quantitation.

### 2.6. Sensory Evaluation

#### 2.6.1. Sample Preparation

The vacuum-packed meat samples were thawed at 4 °C for 24 h prior to cooking. The cooking protocol according to Ma et al. [17] was employed. Thawed samples were heated in a water bath (Model 360, Contherm, New Zealand) at 58–59 °C for 2 h. The meat was then removed from the packaging and then seared using a grill (Breville BGR200BSS Healthsmart Grill, Australia). The grill plate temperature was kept at 180 °C, with the heat equally distributed between bottom and top plates. The lamb pieces (1 cm × 1 cm × 1 cm) were placed on the bottom plate and cooked for 10 s before the top plate was lowered. The samples were then cooked for a further 2 min, and then placed on a plate and left to equilibrate for 2 min prior to serving. This cooking time resulted in samples cooked to an internal temperature of 80 ± 1.5 °C.

#### 2.6.2. Temporal Dominance of Sensations (TDS)

The Auckland University of Technology Ethics Committee approved the sensory study (AUTEC 13/317) carried out in this research. Panellists gave written and informed consent prior to commencement of the study. Sensory testing took place at the Auckland University of Technology Sensory Laboratory, Auckland, New Zealand. Sensory sessions were approximately 60 min in duration and as compensation for their time, participants received supermarket vouchers.

Ten trained panellists (5 males, 5 females) between 21 and 29 years of age participated in this study. Ethics statement and location, selection of participants and their recruitment were carried out similar to that reported by Ma et al. [17]. A similar training approach as reported by Ma et al. [17] was employed. Participants were trained by: 1) provision of five selected sensory attributes that was agreed by participants after discussion; 2) familiarization of panellists with the measurement of sensations using the TDS procedure and use of an unstructured 150 mm line scale, anchored with “None” on the left end and “Extreme” on the right end to rate the individual sensory attribute; 3) carrying out a dummy TDS trial prior to the actual evaluation. This allowed panellists to familiarize with the TDS interface using the FIZZ software v2.46b (Biosystemes, Couternon, France).

Sensory evaluation was carried out on the different cuts of lamb subjected to varying HPP pressure levels and the control. In each session, the panellists (*n* = 10) were presented with five samples, over a total of 27 sessions (15 samples (3 cuts * 5 HPP parameter) in triplicate). TDS measurements were obtained for five sensory attributes (meaty, browned, juicy, livery and oxidized). Each sensory booth was equipped with a computer screen that presented instructions to panellists on how to consume the lamb sample over time. TDS ratings of all samples were obtained using a 150 mm unstructured line scale, with meaty, browned, juicy, livery and oxidized attributes displayed at the top left-hand corner of the computer screen. Panellists were instructed to continuously report any changes in sensory attributes of the lamb sample during a 80-s period using TDS scales; from the first bite to swallowing and, if present, to after taste sensations. The panellist was prompted to place the cooked lamb sample into their mouth and chew the sample at the start. On the 30-s mark, the panellist was asked to swallow the sample. The action of clicking on the line scale that appeared on the computer screen activated the software to record TDS ratings every second for up to 80 s. Detailed instructions were given to standardize consumption behaviour. All ratings were recorded using the FIZZ Acquisition software (FIZZ Network, Biosystemes). A compulsory 45-s break in-between samples was provided to allow panellist to drink and rinse their mouth with filtered water.

The presentation design of samples utilized a balanced position and considered order effects using the Williams Latin Squares design [22]. Products were coded with three-digit random numbers and a 1 cm^3^ portion of cooked meat was served at 60 ± 0.5 °C in a 100 mL cup. Panellists took a compulsory 60-s break in between product evaluation by drinking water and having unsalted crackers. Sensory testing was conducted in a temperature-controlled room (18 °C), under white light in individual booths.

### 2.7. Data Analysis

Mixed-model Analysis of Variance (ANOVA) was carried out on colour, lipid oxidation, fatty acids and free amino acids profiles of processed lamb cuts with differences considered significant at the 5% level, where the main effect of processing, cuts, and their interactions were considered. Additionally, replication and sample effects were considered as covariates. When the main effect of ANOVA was significant, the means were separated by pairwise comparison using the Fisher’s Least Significant Difference test. To further summarise the relationship between HPP processing parameters and different types of cuts, Principal Component Analysis (PCA) was carried out for the fatty acids and free amino acids data sets, and Kaiser Meyer Olkin (KMO) sampling adequacy was calculated. Individual KMO values for fatty acids and amino acids measurements can be found on Appendix A and Appendix A respectively.

Canonical Variate Analysis (CVA) of TDS data was conducted with each sensory dominance durations as the dependent variable to investigate the overall effect of cuts and processing as carried out by Xu et al. [23]. The Hotelling–Lawley Multivariate Analysis of Variance (MANOVA) was also carried out to determine if significant differences between each product loadings at the 5% level existed. The 90% confidence interval ellipses were plotted on the product loadings plot. CI ellipses that overlap indicated that there were no significant differences between the product loadings, while CI ellipses that do not overlap indicated that the products were significantly different.

Finally, Multiple Factor Analysis (MFA) was utilised to summarise the relationship between the chemical and sensory measures. All univariate and multivariate analyses were carried out using RStudio v.1.1.463 (R v.3.5.1; R Development Core Team, Vienna, Austria) and XLSTAT 2018.6 (Addinsoft, NY, USA).

## 3. Results and Discussion

### 3.1. Changes in Colour

The L*, a*, and b* values in lamb meat were measured (Table 1) to investigate the effect of type of cuts and HPP treatments on meat colour. There were no significant differences in L*, a*, and b* values observed between non-HPP treated control shank, loin, and shoulder cuts. L*, a, b*, C*, and h* values were however significantly different with HPP treatment. The L* value of shank, loin and shoulder cuts were significantly higher when treated at 200 MPa, 300 MPa, 400 MPa, and 600 MPa compared to control samples of the corresponding cuts. Similarly, Souza et al. [12] demonstrated that the L* values of three different pork muscles (*M*. *Longissimus*, *M*. *Psoas major*, and *M*. *Triceps brachii*) were significantly higher when HPP level increased from 0 MPa to 215 MPa when held for 15 s. The increase in L* values (“whitening/ brightening”) can be attributed to globin denaturation, heme release, ferrous ion oxidation and changes in water content [24]. Meanwhile, the a* value was significantly lower in shank and shoulder cuts after pressure treatments from 200 to 600 MPa. The a* value of loin cut was only significantly decreased after treatment at 300 MPa. The b* value of shank, loin and shoulder cuts were significantly higher when treated at 300 MPa up to 600 MPa compared to control samples of the corresponding cuts. The increase in b* values due to HPP has been reported to be related to formation of metmyoglobin [11]. Cheftel and Culioli [24] concluded that meat discolouration through pressure processing may result from (1) a whitening effect in the range 200–350 MPa, due to globin denaturation and/or to heme displacement or release, and (2) oxidation of ferrous myoglobin to ferric metmyoglobin, at or above 400 MPa. Bak, Lindahl, Karlsson, and Orlien [25] concluded that myosin was relatively sensitive to pressure at around 180–300 MPa, and will denature to give unclear appearance in cooked meat. Ma and Ledward [9] stated that rare or medium rare steaks or pinkish lamb will not be an option in samples treated at pressures above 200 MPa. Therefore, meat discolouration could be a potential problem when marketing pressurized raw meat, as colour is one of the most vital criteria for consumers when purchasing meat.

### 3.2. Changes in Lipid Oxidation

The effect of HPP treatment on the level of lipid oxidation (TBARS value) in different lamb meat cuts was investigated and results are shown in Table 1. Mixed-model ANOVA results showed that the lipid oxidation of lamb samples was significantly (*p* < 0.05) affected by both HPP treatment and type of cuts as well as their interaction. There were no significant differences in lipid oxidation between control and samples treated at 200 MPa. However, higher levels of pressure (400 MPa and 600 MP) resulted in higher oxidation values in shank, loin and shoulder cuts of lamb. Studies have reported that lipid oxidation increased with increasing HPP pressures applied to turkey meat [6], chicken [26], and pork mince [27]. Similar to our study, Cheah and Ledward [27] found that the rate of lipid oxidation was significantly increased for the samples treated at 300 MPa and above. It was further reported that lamb (*M. pectoralis profundus*) pressurized at 400 and 600 MPa at 60 °C resulted in the highest TBARS values compared to control samples [13]. Increased lipid oxidation after HPP, may be due to conformational changes of haemoproteins, which results in greater exposure of the catalytic heme group to unsaturated fatty acids [28].

### 3.3. Changes in Fatty Acid Profiles

Fatty acid content influences the nutrition, flavour, and texture of meat [29]. The fatty acid composition of shank, loin, and shoulder meat cuts is summarised in Table 2. The dominant fatty acids in all samples included palmitic (C16:0), stearic (C18:0), oleic (C18:1n9c), and linoleic (18:2n6) acids. Lamb meat has been reported to contain more C16:0, C18:0, and C18:1n9 fatty acids [30,31]. Mixed model ANOVA results in our study showed that the lipid oxidation of lamb samples was significantly (*p* < 0.05) affected by HPP treatments and the different cuts, as well as their interactions.

In terms of cuts, the shoulder cut had significantly higher (*p* < 0.05) amount of saturated fatty (SFA) (especially C16:0 and C18:0), monounsaturated fatty acids (MUFA) (C18:1n9c) and polyunsaturated fatty acids (PUFA) (C18:2n6c and C18:3n3) compared to loin cuts. Similarly, Badiani et al. [32] reported that SFA and PUFA were significantly higher in the shoulder (*M. infraspinatus*) compared to loin (*M*. *longissimus*) cuts of beef meat. The PUFA/SFA ratios for lamb are typically 0.1 but can be higher in some muscles [33]. The highest PUFA:SFA value was found in loin cut (0.355) compared to shank (0.221) and shoulder (0.199) cuts in this study. The ratio of omega 3 to omega 6 PUFAs (n6:n3) ratio is important as a high ratio may promote incidences of cancers and coronary heart disease [34].

Total PUFA content was significantly lower in shank and shoulder cuts after HPP treatment at 200 MPa compared to control samples, and significantly lower after HPP treatment at 300 MPa. With respect to individual fatty acids, the significant changes in PUFA were due mainly to changes in C18:1n9 and C18:2n6 content. The decrease in PUFA might be due to the increase in pressure that can increase oxidation level in lipids. PUFAs are not stable, and its oxidative stability is affected by the composition of fatty acids during processing, aging, and retail display [35].

#### Multivariate Analysis of Fatty Acids Profiles

In order to illustrate differences between each treatment and each cut on the basis of individual fatty acids, PCA was carried out to assess the variation in fatty acids from the three lamb cuts treated with HPP at 200, 300, 400, and 600 MPa. The PCA shown in Figure 1 described 73.33% and 11.49% of the total variation in Factor 1 (F1) and Factor 2 (F2), respectively. Shoulder cut samples treated with HPP at 200, 300, and 400 MPa had high positive scores that corresponded to high loadings of fatty acids, while HPP treatment of shoulder cut sample at 600MPa had low negative scores along F1. C18:3n3, C18:3n6, C22:2n6, and C20:5n3 content were significantly higher (*p* < 0.05) in shoulder cut samples treated with HPP at 200, 300 and 400 MP compared to 600MPa treatment (Table 2). Control and all HPP treated loin cut samples corresponded to high negative scores along F1 that corresponded to low loadings of fatty acids and high TBARS values. This finding is supported by results in Table 2 that showed significantly lower SFA, MFA, and PUFA content in control and HPP processed loin cuts compared to shank and shoulder cuts. Shahidi and Zhong [36] demonstrated that PUFA in meat can react with molecular oxygen through a free radical chain mechanism to form fatty acyl hydroperoxides and other primary oxidation products. Pereda, et al. [37] further showed that fatty acids can decrease as a result of fatty acid oxidation and acidification, thereby supporting the decrease in fatty acid content in this study.

### 3.4. Changes in Free Amino Acid Profiles

The effect of HPP treatments at different pressure levels with different cuts of lamb meat on free amino acids content is summarized in Table 3. Significantly higher (*p* < 0.05) total free amino acids content was found in loin compared to shoulder and shank cuts. To date, no studies have reported the changes in free amino acids content of meat with HPP treatments.

In terms of HPP treatment in this study, HPP treatment of shoulder cut had no adverse effect on total free amino acid composition compared to the corresponding control sample. The concentration of total free amino acids was however significantly higher in the shank cut when treated at both 200 MPa and 600 MPa compared to the corresponding control sample. This was mainly due to the increases in Asp, Glu, Phe, Ala, Thr, and Ser content. A higher level of free amino acids could be due to the extent the overall autolytic activity of raw beef meat when HPP was applied at between 100 and 300 MPa (10 min at 25 °C), especially at 100 MPa [14].

The total free amino acids of HPP treated samples were higher than control. In fact, it was the highest with HPP treatment at 600 MPa compared to other HPP treated samples. His, Leu, Lys, Met, and Pro content significantly higher with HPP treatment at 400 MPa, while Ile, Phe, Tyr, and Gly content were significantly higher when HPP treated at 600 MPa. Proteolysis leads to increased amino acid concentration, while amino acid metabolism decreases the concentration of certain amino acids [38].

#### Multivariate Study of Free Amino Acids Profile

In order to illustrate differences between each treatment and each cut on the basis of individual free amino acids compounds, PCA was carried out to assess the variation in free amino acids content of the three lamb cuts treated with HPP at 200 MPa, 300 MPa, 400 MPa, and 600MPa. The PCA shown in Figure 2 described 58.44% and 9.90% of the total variation in factor 1 (F1) and factor 2 (F2), respectively.

Loin and shank cut samples treated with HPP at 300, 400 and 600 MPa had high positive scores that corresponded to high loadings of free amino acids. This is supported by results shown in Table 3 that showed total free amino acids were significantly higher (*p* < 0.05) in loin and shank cuts treated at 300 and 600MPa compared to shoulder cut. Many researchers suggested that processing meat would increase the presence of certain free amino acids by proteolysis [39]. The increase in free amino acids was similarly reported by Ohmori, et al. [14] who suggested that high pressures between 100 and 300 MPa for 10 min at 25 °C increases the overall autolytic activity of raw beef round (inside cut) meat, and leads to a higher concentration of free amino acids. Control samples for all three cuts had high negative scores along F1 that corresponded to significantly lower (*p* < 0.05) total free amino acids in control cuts compared to corresponding HPP treated cuts (Table 3).

### 3.5. Temporal Dominance of Sensations

TDS curves distinguished the effect of HPP pressure and different cuts in terms of dominant sensory attributes of the samples. Figure 3 showed that all samples were dominant in terms of meaty, browned, juicy and livery attributes during the consumption period, and then oxidized thereafter. Meaty was the first dominant sensation in all samples with the dominance rates starting at over 50% dominance rate and then decreasing to below chance level in less than 10 s. Suzuki, et al. [2] suggested that free amino acids have an important role in determining brothy and meaty flavours, and as precursors of meat flavour. Dominance rate of the browned attribute increased from the start of consumption and reached a maximum in the first 3–5 s, then rapidly decreased to below chance level after 10 s of the consumption period. Starting from 11 s, oxidized became the dominant attribute above significance level until the end of consumption (except for Sk-C (1), Sk-200 (2), L-600 (9), and Sd-300 (13)). The attributes of livery and juicy were occasionally above the significant level, and only lasted a few seconds (except Sk600, L-600 and Sd-600). Oxidized and brothy were found to be the most dominant sensory attributes when using TDS to investigate the changes in beef muscles with PEF treatments [18]. However, to date there has been no reported studies that have investigated temporal changes in the sensory characteristics of high-pressure processed meat samples.

Meaty started at around 60% dominance rate in most shank cut (Sk) samples (1-4), and only for Sk-600 (5) at 75 % dominance rate. After 6 s, the dominance of meaty in all Sk samples (1-5) decreased to below chance level. Meaty became significant at 11s-30s and 55s-80s only in Sk-C (1) sample. Brown attribute was above significance level in Sk-C (1), Sk300 (3) and Sk-600 (5) from 3 to 5 s, but was more dominant in Sk300 (3) at 45% dominance rate. Livery was significant in Sk-200 (2) sample from 30 to 36 s. Oxidized was significant in Sk-C (1) from 45 to 55 s and 60 to 70 s. Oxidized was more dominant in HPP treated samples. It was significant from 20 s onwards in Sk-200 (2) reaching 50% dominance rate between 50 and 80 s. Oxidized was significant from 8 to 15 s, 28 to 48 s, 52 to 60 s, and 65 to 80 s in Sk-300 (3), from 8 to 40 s and 50 to 80 s in Sk-400 (4), and from 7 to 15 s and 70 to 80s in Sk-600 (5) samples. However, dominance was not more 45% dominance rate.

In loin cut (L) samples (6-10), meaty started at around 58% dominance rate in L-C(6), L-200(7), L-300(8) and L-600 (10) samples, and at 75% dominance rate in L-400 (9) sample, which decreased to below chance level at 3 s. Meaty attribute was only significant in L-C (6) samples from 40 to 45 s and 75 to 80 s, as well as from 50 to 75 s in L-300 (8). Browned was significant from 2 to 8 s in all samples except for L-400 (9). Juicy was significant only in L-200 (7) from 5 to 8 s. Livery was significant in L-600 (10) from 20 to 30 s. Oxidized was significant in L-200 (7) from 10 to 80 s with a high dominance rate of 50% between 30 and 50 s. It was also dominant in L600 (10) sample from 25 to 40 s at around 45% dominance rate.

Meaty was dominant in most shoulder cut (Sd) samples with 75% dominance rate in Sd-200 (12), Sd-300(13), Sd-400(14), and Sd-600(15) samples, and 62% dominance rate in Sd-C(11), decreasing to below significance level at about 5 s. Browned was significant in Sd-C (11), Sd-200(12), Sd-400 (14), and Sd-600 (15) from 4s to 8s. Juicy and livery were only significantly dominant in Sd-600 (15) cut between 10-18 s and 28-40 s, respectively. Oxidized was more dominant in HPP treated samples. Sd-200 (12) was dominant in oxidized attribute mainly from 9 to 40 s, Sd-300(13) from 27 to 80 s, and Sd-400 (14) from 18 to 60 s. In Sd-600 (15) sample, oxidized was dominant at about 52% dominance rate only from 60 to 80 s.

Canonical Variate Analysis (CVA) has been used in various TDS studies to provide a clearer interpretation of results [17]. Prior to CVA, the sensory attributes in TDS were converted to TDS scores, by considering the duration and intensity of sensory attributes selected during TDS trials [40]. All samples were discriminated by CVA with a high variance for sensory data at 95.45%. Hotelling Lawley MANOVA (F = 15.414; *p* < 0.01) showed significant differences between the samples in terms of the temporal flavour attributes measured by TDS (Figure 4). F1 explained 75.82% of the variance, separating the meat samples in terms of HPP treatments at different pressure levels. Negative scores along F1 corresponded to control samples of all cuts and almost all samples treated with HPP at 200 and 300 MPa (except L-300). It can be seen that control and samples treated at mild pressure levels (200 MPa) were correlated with meaty and juicy, especially in shank and shoulder cuts.

Positive loadings corresponded to samples treated with HPP at 400 and 600 MPa. Samples treated at the highest-pressure levels (600 MPa) were correlated with browned (Sd-600), livery (SK-600) and oxidized (L-600) attributes. In this study, the higher levels of pressure (400 and 600 MPa) resulted in higher oxidation values in shank, loin, and shoulder cuts of lamb (Table 1). In addition, shank cut treated at 600 MPa was correlated with livery. Livery flavour in this study may be attributed to the high free amino acids produced when content when treated at 600 MPa (Table 3). Methionine is an important precursor of sulphur volatile compounds meat that may interact with carbonyl compounds to produce the livery flavour attribute in pork [41], beef [42], and lamb [43] liver. Interestingly, methionine content was significantly higher in 600 MPa treated shank cut in this study (Table 3).

### 3.6. Overall Relationship Between Chemical and Sensory Measures 

Figure 5 illustrates the MFA plot showing the interrelationship between all physicochemical and sensory measures. Interestingly, chemical results were shown to be strongly correlated with sensory qualities. Samples were separated mainly along F1 with 49.93% variance explained. Samples having negative loadings along F1 were mainly the control samples and samples subjected to low HPP pressure (i.e., 200 MPa) treatments, while samples subjected to higher HPP pressures (i.e., 400 and 600 MPa) had positive loadings along F1. Samples that were subject to low HPP pressure treatments had high negative scores along F1 that were correlated with higher fatty acid content, and described as being meaty and juicy. Total fatty acids have been reported to be more likely in influencing tenderness and juiciness of meat. Lipids can trap moisture in muscle, improving juiciness [33].

Samples that were subject to higher-pressure treatments were correlated with high amino acid content. Koutsidis et al. [44] reported that free amino acids, such as leucine, isoleucine, serine, threonine, valine, and phenylalanine contribute to brown/roasted attribute. TBARS values were closely associated with browned, oxidised, and livery sensory qualities. High TBARS value was strongly correlated with the oxidized attribute. In addition, at pressures of 400 MPa and above, the unsaturated lipids in the meat become more susceptible to oxidation, probably due to the release of iron from complexes present in meat (haemosiderin and ferritin) and/or changes in the lipid membrane [9].

## 4. Conclusions

In this study, it is obvious that HPP significantly affected physicochemical properties and sensory characteristics of the three different lamb cuts. Colour, lipid oxidation, fatty acids, and free amino acids were significantly influenced by HPP treatments at different pressure levels. Lamb meat discolouration occurred when HPP was applied at higher-pressure levels of 400MPa and 600MPa. Lipid oxidation similarly increased significantly with pressure increased after 200 or 300 MPa depending on type of cuts compared to the corresponding control cuts. Free amino acids content increased in shank and loin cuts with pressure increase after 200 MPa, and in shoulder cut after 400 MPa. TDS results showed that HPP processing at high-pressure levels were associated with browned, livery, and oxidized flavour. Future studies can investigate the changes in volatile aroma compounds in the HPP treated samples in order to provide a better understanding of flavour formation with increasing pressure levels. Our results confirm that when carrying out HPP processing of meat, it is important to consider the pressure levels applied and the type of meat cuts used to achieve a product with desirable physical, chemical, and sensory characteristics. Further optimisation studies using response surface methodology may be beneficial to determine the optimum HPP processing parameters to obtain HPP processed lamb meat with desirable physicochemical and sensory qualities.

## Figures and Tables

**Figure 1 molecules-25-02665-f001:**
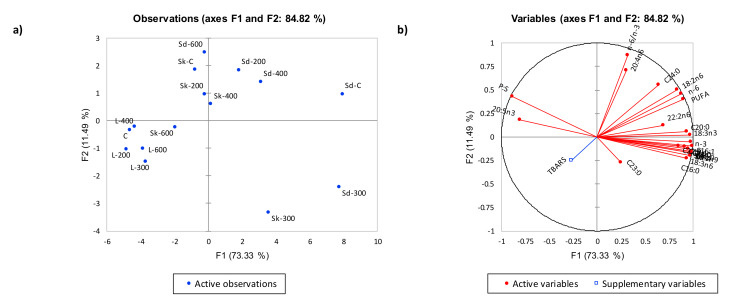
Principal Component Analysis (PCA) plots (**a**) score plot, (**b**) loading plot, for fatty acids content in control and high pressure processing (HPP) treated samples. Red and blue vectors indicate free fatty acids and thiobarbituric acid reactive substances (TBARS) measures, respectively. F1 (73.33%) and F2 (11.49%) explained a total of 84.82% variance. Cuts-Sd: Shoulder, Sk: Shank, L: Loin; HPP-200: 200MPa, 300: 300MPa, 400: 400MPa, 600: 600MPa. Kaiser-Meyer-Olkin (KMO) sampling adequacy: 0.717, detailed KMO values for each fatty acid can be found on Appendix A.

**Figure 2 molecules-25-02665-f002:**
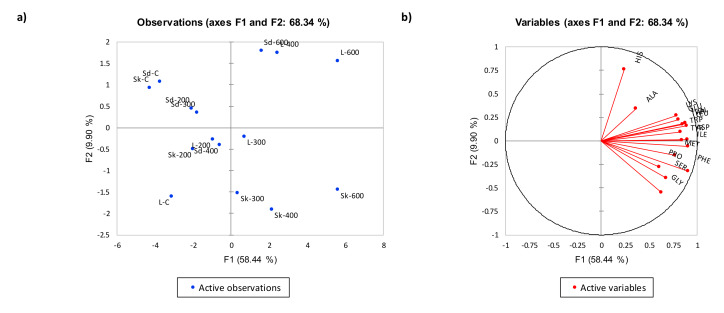
PCA plots (**a**) score plot, (**b**) loading plot, for free amino acids of control and HPP treated samples. Red vectors indicate free amino acids. F1 (58.44%) and F2 (9.90%) explained a total of 68.34% variance. Cuts-Sd: Shoulder, Sk: Shank, L: Loin; HPP-200: 200MPa, 300: 300MPa, 400: 400MPa, 600: 600MPa. Kaiser-Meyer-Olkin sampling adequacy: 0. 773, detailed KMO values for each amino acids can be found on Appendix A.

**Figure 3 molecules-25-02665-f003:**
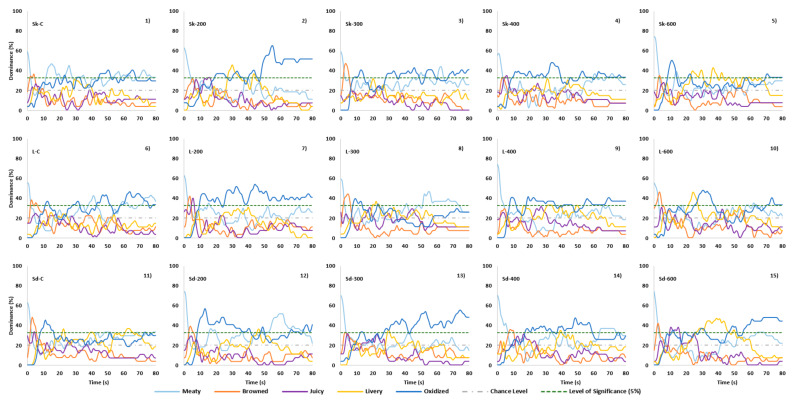
Panel dominance curves for sensory attributes of different lamb cuts with and without HPP treatments for over 80s period. Cuts - Sd: Shoulder, Sk: Shank, L: Loin; HPP – 200: 200MPa, 300: 300MPa, 400: 400MPa, 600: 600MPa; Cuts - Sd: Shoulder, Sk: Shank, L: Loin.

**Figure 4 molecules-25-02665-f004:**
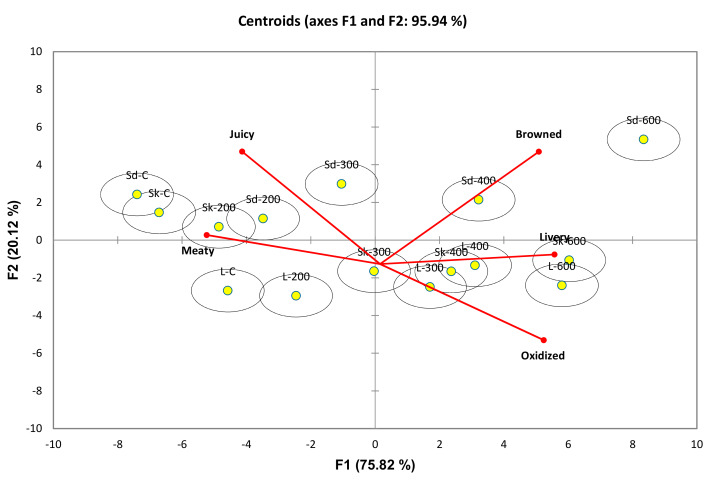
Canonical Variate Analysis (CVA) bi-plot of dominance durations of flavours in control and high pressure processed lamb meats. Hotelling–Lawley MANOVA test showed significant product differences based on flavour attributes of cooked lamb samples. The 90% confidence ellipses were added to indicate statistical significance. Non-overlapping ellipses indicate sample centroid are significantly different, while overlapping ellipses suggest sample centroid are not significant. Cuts-Sd: Shoulder, Sk: Shank, L: Loin; HPP-200: 200MPa, 300: 300MPa, 400: 400MPa, 600: 600MPa.

**Figure 5 molecules-25-02665-f005:**
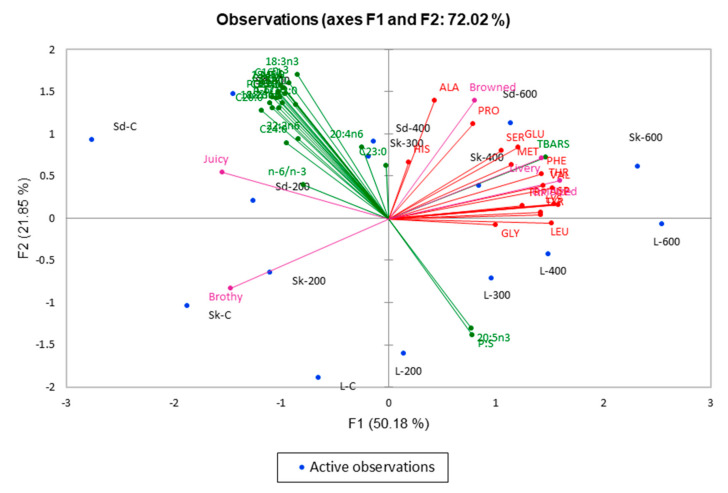
Multiple Factor Analysis (MFA) bi-plot for amino acid (red), free fatty acid (green), and Temporal Dominance of Sensations (TDS) Scores (purple) for samples varying in cuts and HPP treatments. F1 (50.18%) and F2 (21.85%) explained a total of 72.02% variance.

**Table 1 molecules-25-02665-t001:** The L*, a*, b*, h*, and lipid oxidation values of four different lamb cuts with and without HPP treatments at 200, 300, 400, and 600 MPa (mg/100 g dry meat).

Measurements		Cuts	Treatment					F-Value		
			C	200	300	400	600	Cut	Treatment	Cut*Treatment
**Colour**	L	Shank	41.25d	46.845c	53.455b	57.457a	59.258a	165.714	417.769 ***	59.713 ***
		Loin	39.258c	44.745c	51.581b	55.368b	57.254a			
		Shoulder	40.465d	45.465c	52.165b	56.267a	58.236a			
	a	Shank	6.845ay	6.235by	6.325by	6.025b	5.212c	218.134 **	327.524 ***	45.183 ***
		Loin	7.258ax	7.025bx	7.125bx	5.925c	5.845c			
		Shoulder	7.526ax	7.125bx	7.235bx	6.015c	5.825c			
	b	Shank	8.325by	8.125by	14.258a	14.925ax	15.25ax	325.127 **	123.135 ***	79.815 ***
		Loin	9.052bx	8.925bx	14.954a	15.125ax	15.369ax			
		Shoulder	9.358bx	9.423bx	14.136a	14.354ay	14.527ay			
	C	Shank	10.778cy	10.242cy	15.598by	16.095a	16.116a	125.132 **	233.365 ***	52.325 ***
		Loin	11.602bx	11.358bx	16.565ax	16.095a	16.443a			
		Shoulder	12.009bx	11.813bx	15.88ay	16.095a	15.651a			
	h	Shank	50.572c	52.498cx	66.077bx	68.017a	71.131ax	153.142 **	432.325 ***	49.165 ***
		Loin	51.277c	51.793cy	64.524by	68.608a	69.178ay			
		Shoulder	51.193c	52.906cx	64.525by	67.264a	68.15ay			
**TBARS**		Shank	0.188cy	0.206cx	0.314by	0.32by	0.446a	158.274 ***	347.745 ***	69.793 ***
		Loin	0.192dy	0.249cx	0.421bx	0.464ax	0.479a			
		Shoulder	0.212cx	0.185cy	0.338by	0.346by	0.443a			

^a,b,c,d^ means with different letters within a row show the significant effects of processing in each cut; ^x,y,z^ means with different letters within the column show the significant effects of different cuts under the same processing conditions using Fisher’s least significant difference (*p* < 0.05). * *p* < 0.05, ** *p* < 0.01, *** *p* < 0.001. TBARS: Thiobarbituric acid reactive substances.

**Table 2 molecules-25-02665-t002:** Fatty acid composition of different lamb cuts with and without HPP treatments at 200, 300, 400, and 600 MPa (mg/100 g dry meat).

Fatty Acids	Cuts	Treatment					F-Value		
		C	200	300	400	600	Cut	Treatment	Cut*Treatment
**C16:1**	Shank	0.687ax	0.477bx	0.445cy	0.418dy	0.422dx	261.107 ***	433.725 ***	164.281 ***
	Loin	0.315z	0.282z	0.321y	0.314y	0.299y			
	Shoulder	0.77ax	0.787ax	0.576bx	0.53cx	0.473dx			
**C17:1**	Shank	0.387ax	0.333bx	0.275cx	0.267cx	0.272cx	518.053 ***	117.496 ***	60.766 ***
	Loin	0.203ay	0.191aby	0.193aby	0.176by	0.177by			
	Shoulder	0.368ax	0.388ax	0.289bx	0.292bx	0.24cx			
**C18:1n9**	Shank	15.682ax	9.907by	9.373cy	9.028dy	9.154dx	432.277 ***	226.99 ***	379.154 ***
	Loin	7.218ay	7.179az	6.149bz	6.157bz	6.143by			
	Shoulder	17.919ax	17.132bx	12.666cx	11.681dx	9.165ex			
**C18:2n6**	Shank	3.295ay	2.818by	2.773by	3.09ay	2.792by	530.514 ***	214.87 ***	40.229 ***
	Loin	2.496az	2.283by	2.222bcy	2.219bcz	2.17cz			
	Shoulder	4.286ax	3.992bx	3.62cx	3.837cx	3.446ex			
**C18:3n3**	Shank	0.943ax	0.91ax	0.77by	0.729by	0.759by	111.142 ***	112.323 ***	56.296 ***
	Loin	0.676z	0.677y	0.677z	0.632y	0.628y			
	Shoulder	1.064ax	0.933bx	1.099ax	0.966bx	0.826cx			
**C18:3n6**	Shank	0.234y	0.291x	0.359y	0.252x	0.247x	131.136 ***	30.547 ***	17.136 ***
	Loin	0.18ay	0.184ay	0.184az	0.153by	0.151by			
	Shoulder	0.386ax	0.26bx	0.407ax	0.264bx	0.182cy			
**C22:2n6**	Shank	0.187ay	0.174abx	0.182a	0.177aby	0.122by	62.098 ***	26.752 ***	33.093 ***
	Loin	0.188ay	0.167bx	0.17b	0.145cz	0.133dz			
	Shoulder	0.308ax	0.142cy	0.199b	0.219bx	0.177bcx			
**C20:4n6**	Shank	0.447ax	0.408bx	0.388bcy	0.383bcx	0.353c	165.584 ***	57.394 ***	15.397 ***
	Loin	0.389ay	0.327by	0.343by	0.332by	0.335b			
	Shoulder	0.434ax	0.416ax	0.421ax	0.379bx	0.368b			
**C20:5n3**	Shank	0.341x	0.32	0.302	0.339x	0.314x	73.368 ***	17.449 ***	8.197 ***
	Loin	0.368ax	0.319b	0.329b	0.33bx	0.326bx			
	Shoulder	0.315ay	0.299ab	0.304ab	0.28bcy	0.254cy			
**C16:0**	Shank	9.688ay	6.118by	5.607cy	5.721cy	5.273dx	526.279 ***	159.121 ***	681.42 ***
	Loin	4.208z	4.035z	4.697y	4.15cy	4.624y			
	Shoulder	11.398ax	11.088ax	7.015bx	7.498bx	5.329cx			
**C17:0**	Shank	0.603ax	0.39by	0.369by	0.418bx	0.361bx	147.218 ***	350.095 ***	116.525 ***
	Loin	0.285z	0.268z	0.317y	0.283y	0.297y			
	Shoulder	0.772ax	0.722bx	0.48cx	0.515cx	0.397dx			
**C18:0**	Shank	8.229ay	5.714bx	5.021cy	5.179cy	4.505dy	411.091 ***	192.849 ***	615.999 ***
	Loin	4.295az	4.225ay	3.928bz	3.544cz	3.629cz			
	Shoulder	14.063ax	13.009bx	9.102cx	6.499dx	5.508ex			
**C20:0**	Shank	0.374ay	0.33by	0.283cy	0.27cdy	0.258dy	193.28 ***	95.328 ***	64.125 ***
	Loin	0.244az	0.243az	0.238aby	0.231by	0.231by			
	Shoulder	0.422ax	0.434ax	0.359bx	0.338bx	0.288cx			
**C21:0**	Shank	5.076ay	4.427bx	3.321cy	3.366cy	2.609dy	125.427 ***	336.28 ***	142.889 ***
	Loin	2.5abz	2.535ay	2.499aby	2.402bz	2.242cy			
	Shoulder	6.819ax	5.441bx	4.839cx	4.327dx	3.183ex			
**C22:0**	Shank	0.209y	0.202y	0.212y	0.204x	0.189	40.828 ***	15.461 ***	3.594 **
	Loin	0.183y	0.187y	0.207y	0.189y	0.185			
	Shoulder	0.247ax	0.222bx	0.224bx	0.209bx	0.193c			
**C23:0**	Shank	0.167ax	0.13abx	0.105b	0.095b	0.099b	6.995 **	6.363 **	18.143 ***
	Loin	0.136ay	0.105by	0.091c	0.095c	0.095c			
	Shoulder	0.109by	0.154ax	0.109b	0.1b	0.098b			
**C24** **:0**	Shank	0.358x	0.337x	0.314	0.347	0.298	49.74 ***	12.156 ***	8.197 ***
	Loin	0.274y	0.276y	0.283	0.317	0.287			
	Shoulder	0.407ax	0.326bx	0.308b	0.329b	0.338b			
**SFA**	Shank	24.729ay	17.673by	15.258cy	15.625cy	13.617dy	103.453 ***	236.456 ***	479.597 ***
	Loin	12.149az	11.90ay	12.285ay	11.236by	11.615bz			
	Shoulder	34.261ax	31.421bx	22.461cx	19.841cx	15.359dx			
**MFA**	Shank	16.766ay	10.727by	10.103bx	9.723bx	9.857bx	594.525 ***	570.443 ***	317.29 ***
	Loin	7.745az	7.662az	6.673by	6.656by	6.628by			
	Shoulder	19.067ax	18.316ax	13.54bx	12.512bx	9.888cx			
**PUFA**	Shank	5.466ay	4.94by	4.793by	4.989by	4.606cy	300.643 ***	118.681 ***	42.989 ***
	Loin	4.316az	3.975az	3.944az	3.8305bz	3.762bz			
	Shoulder	6.811ax	6.061bx	6.068bx	5.964bx	5.272cx			
**P:S**	Shank	0.221cy	0.28cy	0.314b	0.319by	0.338a	346.12 ***	244.411 ***	84.817 ***
	Loin	0.355x	0.334x	0.321	0.341x	0.324			
	Shoulder	0.199by	0.193bz	0.27a	0.301ay	0.343a			
**n-3**	Shank	1.291ay	1.236ax	1.078by	1.074by	1.08	605.587 ***	69.808 ***	38.893 ***
	Loin	1.05y	1.002y	1.012y	0.969y	0.96			
	Shoulder	1.385ax	1.239bx	1.409ax	1.253bx	1.086b			
**n-6**	Shank	4.175ay	3.703by	3.715by	3.914ax	3.526bx	375.356 ***	137.988 ***	34.198 ***
	Loin	3.265y	2.973z	2.932bz	2.862y	2.802y			
	Shoulder	5.426ax	4.822bx	4.659bx	4.711bx	4.186cx			
**n-6/n-3**	Shank	3.237by	2.995by	3.446ax	3.644ay	3.266by	406.55 ***	155.894 ***	32.015 ***
	Loin	3.11y	2.969y	2.897y	2.955x	2.918y			
	Shoulder	3.919ax	3.894ax	3.308bx	3.761ax	3.855ax			
**Total**	Shank	46.96ay	33.339by	30.154cy	30.336cy	28.079cy	288.075 ***	366.051 ***	162.589 ***
	Loin	24.21z	23.536z	22.901z	21.722z	22.005z			
	Shoulder	60.139ax	55.798bx	42.069cx	38.317cx	30.519dx			

^a,b,c,d^ means with different letters within a row show the significant effects of processing in each cut; ^x,y,z^ means with different letters within the column show the significant effects of different cuts under the same processing conditions using Fisher’s least significant difference (*p* < 0.05). * *p* < 0.05, ** *p* < 0.01, *** *p* < 0.001. Saturated Fatty Acids (SFA) = C14:0+C16:0+C17:0+C18:0+C20:0+C21:0+C22:0+C24:0. Monounsaturated Fatty Acids (MUFA) = C16:1+C18:1 cis-9+C20:1 cis-9. Polyunsaturated Fatty Acids (PUFA) = C18:2n6+C18:3n3+C20:2n6+C20:3n6+C20:5n3. n−6 = C18:2n6+C20:2n6+C20:3n6. n−3 = C18:3n3+C20:5n3.

**Table 3 molecules-25-02665-t003:** The free amino acid composition of different lamb cuts with and without HPP treatments at 200, 300, 400, and 600 MPa (mg/100 g dry meat).

Free Amino Acids		Cuts	Treatment					F-Value		
			C	200	300	400	600	Cut	Treatment	Cut*Treatment
**Essential Amino Acids**	His	Shank	0.15	0.143	0.143	0.124y	0.136y	0.485	2.681	4.137 **
		Loin	0.114b	0.12b	0.123b	0.168ax	0.161ax			
		Shoulder	0.144	0.147	0.143	0.126y	0.153y			
	Ile	Shank	0.277ax	0.199bx	0.114cy	0.117cy	0.148bcy	6.926 **	5.964 **	7.725 ***
		Loin	0.168by	0.174bx	0.187abx	0.208abx	0.254ax			
		Shoulder	0.127by	0.125by	0.124by	0.2ax	0.196ay			
	Leu	Shank	0.186ax	0.187ax	0.163aby	0.134by	0.157aby	18.729 ***	7.274 **	6.977 ***
		Loin	0.149cy	0.168bcx	0.184bcx	0.197bx	0.245ax			
		Shoulder	0.137cz	0.153bcy	0.145bcz	0.17ax	0.155aby			
	Lys	Shank	0.124ax	0.108aby	0.094bc	0.084cy	0.103abcy	7.828 **	3.808 *	6.379 ***
		Loin	0.107bx	0.109by	0.093b	0.15ax	0.143ax			
		Shoulder	0.083cy	0.128ax	0.091bc	0.094bcy	0.112aby			
	Met	Shank	0.065bx	0.059b	0.054bc	0.037cy	0.143ax	8.059 **	14.134 ***	18.044 ***
		Loin	0.044by	0.047b	0.049b	0.079ax	0.074ay			
		Shoulder	0.048aby	0.045b	0.061ab	0.065ax	0.064aby			
	Phe	Shank	0.171c	0.216abx	0.18bc	0.124dy	0.237ax	3.723 *	5.543 **	5.286
		Loin	0.17b	0.159by	0.174ab	0.18abx	0.215ax			
		Shoulder	0.157ab	0.171aby	0.169ab	0.142bxy	0.183axy			
	Thr	Shank	0.145cdy	0.252ax	0.228abx	0.133dy	0.186bcy	7.352 **	6.608 **	6.485 ***
		Loin	0.138by	0.257ax	0.231ax	0.247ax	0.277ax			
		Shoulder	0.204abcy	0.162cy	0.169bcy	0.211abx	0.234ax			
	Val	Shank	0.231abx	0.268ax	0.248abx	0.175by	0.26ay	4.566 *	3.424 *	6.996 ***
		Loin	0.18cy	0.218bcy	0.272abx	0.285abx	0.329ax			
		Shoulder	0.18cdy	0.204bcy	0.163dy	0.225bx	0.308ax			
**Non-Essential Amino Acids**	Asp	Shank	0.824dy	1.382b	1.113cy	1.061cdy	2.107ax	7.827 **	6.845 **	12.028 ***
		Loin	1.077by	1.294ab	1.523ax	1.543ax	1.669ax			
		Shoulder	1.531ax	1.262ab	0.99by	0.978by	0.999by			
	Glu	Shank	0.808bx	1.192ax	0.61cy	0.554cy	0.854by	1.645	19.974 ***	28.402 ***
		Loin	0.19cy	0.735by	1.134ax	1.017ax	1.211ax			
		Shoulder	0.675cx	0.753bcy	0.818abx	0.921ax	0.829aby			
	Trp	Shank	0.082ax	0.064ab	0.065ab	0.058by	0.075aby	8.672 **	5.178 **	4.972 **
		Loin	0.054dy	0.06cd	0.073bc	0.075bx	0.098ax			
		Shoulder	0.06y	0.06	0.065	0.054y	0.058y			
	Tyr	Shank	0.203ax	0.132bc	0.162bx	0.093dx	0.107cdx	15.139 ***	28.381 ***	7.151 ***
		Loin	0.117by	0.157b	0.138by	0.109bx	0.265ax			
		Shoulder	0.109by	0.107b	0.173ax	0.077cy	0.095by			
	Ala	Shank	0.752bx	1.046ax	0.846ab	0.64b	0.704by	21.416 ***	6.234 **	5.894 **
		Loin	0.543cy	0.579bcy	0.732ab	0.751ab	0.784ay			
		Shoulder	0.84bx	0.976bx	0.844b	0.801b	1.314ax			
	Gly	Shank	0.734bx	0.813abx	0.75abx	0.562cx	0.899ax	42.08 ***	6.412 **	2.919 *
		Loin	0.531by	0.613aby	0.61aby	0.588abx	0.703ax			
		Shoulder	0.504ay	0.564ay	0.505az	0.332by	0.497ay			
	Pro	Shank	0.099abx	0.123ax	0.126ax	0.06by	0.108a	7.249 **	2.18	3.692 *
		Loin	0.064by	0.053by	0.068by	0.095ax	0.098a			
		Shoulder	0.071y	0.085y	0.075y	0.099x	0.102			
	Ser	Shank	0.141by	0.321ax	0.322ax	0.143by	0.304a	0.625	5.416 **	4.558 **
		Loin	0.218x	0.196y	0.256y	0.264x	0.243			
		Shoulder	0.186xy	0.242xy	0.221y	0.245x	0.241			
**Total**		Shank	4.993bx	6.503ax	5.218by	4.1cy	6.529ax	3.864 *	6.774 **	10.353 ***
		Loin	3.864cy	4.939bcy	5.847bx	5.956bx	6.769ax			
		Shoulder	5.056abx	5.185aby	4.758bz	4.739by	5.538ay			

^a,b^ means with different letters in row show a significant effect of processing in each cut; ^x,y,^ means with different letters in column show a significant effect of a cut in each processing using Fisher’s least significant difference (*p* < 0.05).ASP: aspartic acid. GlLU: glutamic acid. GLY: glycine. HIS: histidine. ILE: isoleucine. LEU: leucine. LYS: lysine. MET: methionine. PHE: phenylalanine. PRO: proline. SER: serine. THR: threonine. TRP: tryptophan. TYR: tyrosine. VAL: valine.* *p* < 0.05, ** *p* < 0.01, *** *p* < 0.001.

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
