# Peer review of "The Impact of High-Pressure Processing on Physicochemical Properties and Sensory Characteristics of Three Different Lamb Meat Cuts"

_molecules, 2020, doi:10.3390/molecules25112665_

Round 1
Reviewer 1 Report
Ms. Ref. No. Molecules-736340
Title The impact of high-pressure processing on physicochemical properties and sensory
characteristics of three lamb meat cuts
Authors Ma et al.
Journal Molecules
GENERAL REVIEW
This paper describes a study where the effects of high-pressure processing (200-600 MPa for 1 min) on the physicochemical properties and sensory characteristics of different lamb meat cuts were investigated.
Overall, the paper was easy to read and it is evident the authors tried to describe the physicochemical properties and sensory characteristics of HPP treated lamb meats. I think the data they have collected are valuable, but they are not readily applicable to the HPP industry because industry uses colder temperatures, higher pressurization rates, and longer dwell times than those used in this study.
DETAILED REVIEW
Abstract
No comment
Introduction
1. Line 41 – Replace “mild technology” with “nonthermal process”.
2. Lines 70-85 – I understand the authors are providing an explanation on TDS, but discussion should
be limited to TDS only…not PEF.
Materials and Methods
3. Line 96 – Replace “done” with “conducted”.
4. Lines 104-111 – Thank you for providing information regarding processing conditions. I think authors should be cautious about making conclusions about their results being directly applicable to the meat and HPP industries for the following reasons:
a. Typically, processing fluid temperatures are 4-5 deg C in industrial HPP applications, especially with meat samples since temperatures can cause discoloration quickly. The temperatures used in the study (7-8 deg C) are on the “warm” side, so I would caution the authors from attributing any discoloration to pressure only…the initial product temperature and subsequent adiabatic heating would contribute to any discoloration, as well.
b. Authors should be aware that a pressurization rate of 125 MPa per min is relatively slow. In industry, pressurization rates of 250 to 450 MPa per min are applied. So, in this study, the meat samples are actually under pressure for longer than a typical 1 min dwell time one
would see in industry. Also, in industry, when HPP is applied to meat samples, dwell times often range from 3-5 min…and, for high fat samples, dwell times can be as high as 7-10 min. The long dwell times are necessary to inactivate pathogens and spoilage microorganisms. A dwell time of 1 min, as used in this study, is not economically viable or practiced in industry.
c. The statement, “The temperature reached after pressure build up was less than 25 deg C.” Generally speaking, one would expect a 3 deg C rise for every 100 MPa applied. So, when apply 600 MPa (the highest pressure tested in the study), one would expect temperature to rise by 18 deg C…so the temperature reached after pressure build up should be 7 deg C plus
18 deg C = 25 deg C. So, this statement is correct. However, I don’t understand the following statement, “Water temperature after pressurization were 14°C, 21°C, 28°C, and 42°C, respectively.” – which corresponds to applied pressures of 200, 300, 400, and 600 MPa. There’s something wrong here. The two statements provided by the authors are contradictory and the reported water temperatures after pressurization are too high. Can the authors provide an explanation regarding these high temperatures observed?
Given these discrepancies between typical HPP conditions in industry vs. HPP conditions used in this study, I ask authors to recognize this difference and exercise care about extrapolating their results to industry applications.
5. Line 123 – Please remove decimal place in “14 000 rpm”. For some readers (US mostly), this would read 14 rpm instead of 14 thousand rpm.
Results and Discussion –
6. Lines 244-247 – Two statements discussing the “brightening” or “lightening” effect observed with increasing pressure are repetitive. Resolve to just one sentence.
7. Line 249 – Misspelled “decreased”.
8. Tables 1 and 2 headings have inconsistent formats.
9. Lines 322 and 375 – Multivariate analysis (specifically, PCA) should be described in “Data Analysis” in Materials and Methods.
Conclusions
10. I agree with the authors’ conclusions – “Our results confirm that when carrying out HPP processing of meat, it is important to consider the pressure levels applied and the type of meat cuts used to achieve a product with desirable physical, chemical and sensory characteristics. Further optimization studies using response surface methodology may be beneficial to determine the optimum HPP processing parameters based on different sample cuts and pressure levels applied.” Authors should consider using processing conditions that HPP industry uses in their next optimization study.
References
No comment
Figures
No comment
Author Response
GENERAL REVIEW
This paper describes a study where the effects of high-pressure processing (200-600 MPa for 1 min) on the physicochemical properties and sensory characteristics of different lamb meat cuts were investigated.
Overall, the paper was easy to read and it is evident the authors tried to describe the physicochemical properties and sensory characteristics of HPP treated lamb meats. I think the data they have collected are valuable, but they are not readily applicable to the HPP industry because industry uses colder temperatures, higher pressurization rates, and longer dwell times than those used in this study.
DETAILED REVIEW
Abstract
No comment
Introduction
- Line 41 – Replace “mild technology” with “nonthermal process”.
The word has been replaced accordingly
- Lines 70-85 – I understand the authors are providing an explanation on TDS, but discussion should be limited to TDS only…not PEF.
The authors are trying to introduce the TDS method which is reflected on line 70-78, and expanded it to line 79-85 to the first author’s previous study that employed TDS on meat treated with PEF samples.
Materials and Methods
- Line 96 – Replace “done” with “conducted”.
The word has been replaced accordingly
- Lines 104-111 – Thank you for providing information regarding processing conditions. I think authors should be cautious about making conclusions about their results being directly applicable to the meat and HPP industries for the following reasons:
a. Typically, processing fluid temperatures are 4-5 deg C in industrial HPP applications, especially with meat samples since temperatures can cause discoloration quickly. The temperatures used in the study (7-8 deg C) are on the “warm” side, so I would caution the authors from attributing any discoloration to pressure only…the initial product temperature and subsequent adiabatic heating would contribute to any discoloration, as well.
This is a very fair comment by the reviewer, we have incorporated a statement re to this in our Conclusion as a limitation. The statement reads:
“It is however important to note that the temperature used in this study was higher (7-8oC) compared to processing fluid temperatures in industrial HPP application (4-5oC) as such higher initial product temperature and its subsequent adiabatic heating would perhaps also contribute to discoloration.”
b. Authors should be aware that a pressurization rate of 125 MPa per min is relatively slow. In industry, pressurization rates of 250 to 450 MPa per min are applied. So, in this study, the meat samples are actually under pressure for longer than a typical 1 min dwell time one would see in industry. Also, in industry, when HPP is applied to meat samples, dwell times often range from 3-5 min…and, for high fat samples, dwell times can be as high as 7-10 min. The long dwell times are necessary to inactivate pathogens and spoilage microorganisms. A dwell time of 1 min, as used in this study, is not economically viable or practiced in industry.
We agree with the reviewer on this statement and have incorporated this as our limitation in the Conclusion section together with the aforementioned comment.
“In addition, pressurisation rate of this study was relatively slow (125 MPa per min) compared to industrial practice (250-450MPa per min) with a shorter dwelling time of 1 minute compared to industry standard of 3-10 minutes dependent on the sample. ”
c. The statement, “The temperature reached after pressure build up was less than 25 deg C.” Generally speaking, one would expect a 3 deg C rise for every 100 MPa applied. So, when apply 600 MPa (the highest pressure tested in the study), one would expect temperature to rise by 18 deg C…so the temperature reached after pressure build up should be 7 deg C plus 18 deg C = 25 deg C. So, this statement is correct. However, I don’t understand the following statement, “Water temperature after pressurization were 14°C, 21°C, 28°C, and 42°C, respectively.” – which corresponds to applied pressures of 200, 300, 400, and 600 MPa. There’s something wrong here. The two statements provided by the authors are contradictory and the reported water temperatures after pressurization are too high. Can the authors provide an explanation regarding these high temperatures observed?
We have noticed that this is a mistake especially re water temperature measured, this statement has been deleted.
Given these discrepancies between typical HPP conditions in industry vs. HPP conditions used in this study, I ask authors to recognize this difference and exercise care about extrapolating their results to industry applications.
We agree with the reviewer, we have added this as a limitation in our conclusion and has reworded our statements re industry applications.
- Line 123 – Please remove decimal place in “14 000 rpm”. For some readers (US mostly), this would read 14 rpm instead of 14 thousand rpm.
We have deleted the decimal place for consistency
Results and Discussion –
- Lines 244-247 – Two statements discussing the “brightening” or “lightening” effect observed with increasing pressure are repetitive. Resolve to just one sentence.
We have deleted the latter statement to reduce redundancy.
- Line 249 – Misspelled “decreased”.
Fixed accordingly
- Tables 1 and 2 headings have inconsistent formats.
We have aligned the heading correctly.
- Lines 322 and 375 – Multivariate analysis (specifically, PCA) should be described in “Data Analysis” in Materials and Methods.
We have added a statement on PCA under Data Analysis section
“In addition, Principal Component Analysis was utilised to project the overall relationship of the response (e.g. free fatty acids) relative to the product (e.g. meat cuts)”
Conclusions
- I agree with the authors’ conclusions – “Our results confirm that when carrying out HPP processing of meat, it is important to consider the pressure levels applied and the type of meat cuts used to achieve a product with desirable physical, chemical and sensory characteristics. Further optimization studies using response surface methodology may be beneficial to determine the optimum HPP processing parameters based on different sample cuts and pressure levels applied.” Authors should consider using processing conditions that HPP industry uses in their next optimization study.
As aforementioned as well that we will also incorporate the limitations of the study in the Conclusion section.

Reviewer 2 Report
Review of the paper molecules-736340 submitted for publication in Molecules journal.
The paper by Ma and co-workers investigated the effects of high-pressure processing on the physicochemical properties and sensory characteristics of lamb meat cuts. The topic is not novel in Meat Science, but the experimental design sounds good. The authors detailed several aspects that are not needed and some information is not relevant and it is worthy to consider them as supplementary data or remove. The discussion is very very long. Please try to be synthetic and do a short discussion but objective. Also, I have a MAJOR comment about the principal component analyses. They are ALL presented wrongly. See my comment below.
First of all, the introduction has to be shortened by half. It is very long and the main message is diluted. I ask the authors to identify the novelty of this work and highlight the main objectives about molecules, to feat in the scope of the journal. For example, the sentence in lines 47-48 is not needed. Several examples like this in the whole manuscript. Please revise the manuscript carefully.
Line 91, section M&M. The authors didn’t describe the experimental design concerning the animals, feeding and other information related to breed and gender. Line 94: different lambs means that different effects. The authors have to consider this point very carefully.
Line 111/ how the samples were frozen? Please detail all the procedure used to be able to reproduce your work.
Line 113: Figure 1 is not relevant. Please remove.
Section 2.2, color measurement is not well described. Please detail the protocol. I further recommend to the authors to compute Chroma (C*) and hue angle (h*). The authors can refer for example to this study in Meat Science journal to compute those parameters relevant for the industry: Gagaoua M., Picard B. & Monteils V. (2018) Associations among animal, carcass, muscle characteristics, and fresh meat color traits in Charolais cattle. Meat Sci 140, 145-56. https://doi.org/10.1016/j.meatsci.2018.03.004
Line 231: please check the table 2 and add the color parameters.
The PCAs presented in Figures 1 & 2 need corrections. The eigenvectors cannot be higher than 1. The variables are projected in the two first components with max values of +1 to -1. Please consider this very carefully. Please describe in details how the PCAs analyses were done. Did the authors considered the Measure of Sampling Adequacy? If yes, please give it in the legend of the figures.
I ask also to the authors describe very well their figures or tables by giving all the needed information.
The figure 3 is hard to understand and please check the colors. I ask the authors to find a manner how this can be well described.
The figure 4 is not clear. What about the other variables which are not in blue color? Are they supplementary ?? please check all these aspects before a scientific review can be done. In this same figure, we can see twice the term “observations” please check and correct the figure and make it easy for the reader. Describe all the details in the legend.
Author Response
Please see attached for the review response.

Round 2
Reviewer 2 Report
The referee recommend to separate the loadings from the biplot. The PCA stills false in the present form. The authors presented the Kaiser Meyer Olkin sampling adequacy, which is good but they didn't consider the individuals KMO's scores to remove the variables that are not eligible in the PCA. Please, check and show the raw data in supplementary file.
For color measurements, the authors have to add a reference as previously suggested. Please, for each protocol check that a reference is cited.
The discussion is poor and needed further citations and impovement.
Figure 3 is not mondatory, please move as supplementary data.
Author Response
See attached document for review's replies.
